# How to Achieve Compliance with GDPR Article 17 in a Hybrid Cloud Environment

**Miriam Kelly [1], Eoghan Furey [1] and Kevin Curran [2,*]** 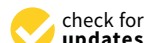

[1] Computer Science Department, Letterkenny Institute of Technology, Letterkenny, Ireland; phkjc@outlook.com (M.K.); eoghanfurey@gmail.com (E.F.)
[2] School of Computing, Engineering & Intelligent Systems, Ulster University, Londonderry, Ireland
* Correspondence: kj.curran@ulster.ac.uk

**Abstract:** On 25 May 2018, the General Data Protection Regulation (GDPR) Article 17, the Right to Erasure ("Right to be Forgotten") came into force, making it vital for organisations to identify, locate and delete all Personally Identifiable Information (PII) where a valid request is received from a data subject to erase their PII and the contractual period has expired. This must be done without undue delay and the organisation must be able to demonstrate that reasonable measures were taken. Failure to comply may incur significant fines, not to mention impact to reputation. Many organisations do not understand their data, and the complexity of a hybrid cloud infrastructure means they do not have the resources to undertake this task. The variety of available tools are quite often unsuitable as they involve restructuring so there is one centralised data repository. This research aims to demonstrate that compliance with GDPR's Article 17 Right to Erasure ("Right to be Forgotten") is achievable in a hybrid cloud environment by following a list of recommendations. However, full retrieval, all of the time will not be possible, but we show that small organisations running an ad-hoc hybrid cloud environment can demonstrate that reasonable measures were taken to be Right to Erasure ("Right to be Forgotten") compliant.

**Keywords:** privacy; general data protection regulation; security; cloud security



## 1. Introduction

The new General Data Protection Regulation (GDPR) came into force on 25 May 2018, replacing the existing data protection framework. Ireland's Data Protection Commissioner, Helen Dixon, has publicly stated that GDPR improves the rights for data subjects by awarding them control over their Personally Identifiable Information (PII) [1]. This new regulation also imposes strict obligations for data controllers and data processors, who subsequently may incur significant fines of up to EUR 20 million if they cannot demonstrate compliance. In recent years, many small organisations have become dependent on a hybrid cloud environment that they haphazardly implemented as a solution to meet their business needs. Based on the popularity and wide-spread adoption of these solutions, the hybrid cloud market is expected to increase [2]. No two hybrid clouds are alike, and few standards exist thus presenting even further challenges. Introduction of the new GDPR Article 17 legislation which awards individuals the right to request the removal of their data from third party systems and storage imposes a variety of burdensome tasks upon small organisations, requiring them to rethink and modify how they manage Personally Identifiable Information (PII). Many organisations are only processing and using a fraction of the data they store, and therefore clearly do not understand their data [2]. This can be due to sprawling legacy systems, siloed databases, and sporadic automation. PII is a very valuable commodity for hackers, despite this many small organisations often mistakenly believe they have nothing worth stealing or that they are too small to gain a hacker's attention. Consequently, investing in security is a low priority, making them easy targets [3]. However, 43 percent of cyber-attacks target small business and 55 percent of attacks come

from within the organisation itself [4]; some of these may be categorised as malicious, but many are simply attributed to innocent user mistakes. Therefore, even with the best security in place, if privacy policies are not enforced, PII can still be accessed.

Organisations must understand the PII they are responsible for and be able to identify and locate all PII when the contractual agreement that allows them to possess and process the data is to expire. This is inclusive of all PII retained for a data subject when a valid request to erase the data has been received, so they can review and delete the same, without undue delay. PII is not necessarily just stored in databases, it may be retained in various formats, in a variety of internal and external locations throughout an organisation's infrastructure. The complexity of the hybrid cloud environment also makes the implementation of security more difficult, as there is now more than one environment to secure. Under the GDPR's Article 17, organisations must be capable of demonstrating that they have taken reasonable measures to be compliant with the Right to Erasure legislation. With the introduction of Article 17, Right to Erasure ("Right to Be Forgotten"), it is crucial that organisations understand their PII. Right to Erasure ("Right to Be Forgotten") enhances the rights of data subjects, so it is vital an organisation can identify and locate PII, both for a data subject where a valid request has been received, and for PII where the contract has expired as this PII must be erased. If either of these are not carried out, without undue delay, the organisation may face significant fines, not only payable to the supervisory authority, but also payable to the data subjects that were put at risk, who may or may not be existing clients. Even organisations that already use industry standard best practices like ISO/IEC 27001, ISO/IEC 27,002, ISO/IEC 17,788, ISO/IEC 17,789, PCI DSS, OWASP, COBIT, ITIL will also need to do a complete review of their data processing as GDPR has broadened the scope of PII, so they must ensure that they are still fully compliant. There are a variety tools available to identify and locate PII, with most involving a centralised visual management point. These tools, however, may not be feasible for a small organisation, as, on top of the cost, they most likely would also involve a major overhaul to the structure of the organisation. Whilst it may not be possible to locate all PII all the time, it is imperative that an organisation can demonstrate that it has taken reasonable measures to be Right to Erasure ("Right to Be Forgotten") compliant and therefore avoid penalties. Compliance must come from the top and it is recommended that organisational policies are put in place to cater for the privacy of PII and anomalies like hard copies and data stored on removable devices, phones, etc. Bearing in mind even if a Payment Card Industry Data Security Standard (PCI DSS) framework, and an International Organisation for Standardization (ISO) 27001 Information Security Management System (ISMS), has been properly implemented, whilst this can offer a good starting point for organisations in becoming GDPR, Right to Erasure ("Right to Be Forgotten") compliant, mistakes can, however, occur if privacy policies and procedures are not enforced.

This research offers insight into the challenges a small organisation may face when trying to identify and locate PII within a hybrid cloud, as it is not just one environment; they are dealing with separate entities. We test how best to identify, locate and report PII stored in a variety of data formats and locations within an experimental hybrid cloud environment for a small organisation, and investigate the challenges, with a view to proposing a set of practical guidelines a small organisation can use to demonstrate reasonable measures were taken for Right to Erasure ("Right to Be Forgotten") compliancy. We focus on small organisations using a hybrid cloud infrastructure, who have little understanding if any, of what constitutes PII and where this PII is stored. As many small organisations do not have the resources and technical expertise required to identify and locate this data, this highlights the question of the challenges a small organisation may face while implementing the GDPR Article 17 "Right to Erasure" within a hybrid cloud storage environment. We demonstrate that if simple guidelines and recommendations are adhered to, compliance with the GDPR Article 17 "Right to Erasure" is achievable in a hybrid cloud environment. The objective is to propose a set of practical guidelines that a small organisation utilizing a hybrid cloud environment can use to demonstrate that reasonable measures were taken to become Right

to Erasure ("Right to be Forgotten") compliant and demonstrate that it is able to identify, locate and report the location of PII for a specific data subject upon receiving a valid request and where the contractual date is due to expire.

## 2. General Data Protection Regulation (GDPR)

GDPR automatically became law in each member state without the need for local implementation, aiming to modernise and harmonise data privacy laws between the Member States, and introduced one legal framework to improve enforcement and reduce costs for organisations, hopefully encouraging economic growth across Europe [5]. GDPR also aims to improve and expand the rights for data subjects, giving them control over the collection and processing of their personal data [1]. GDPR contains 99 Articles, which are challenging for small organisations to understand and become compliant [5]. A brief overview of the key elements of GDPR are shown in Figure 1.

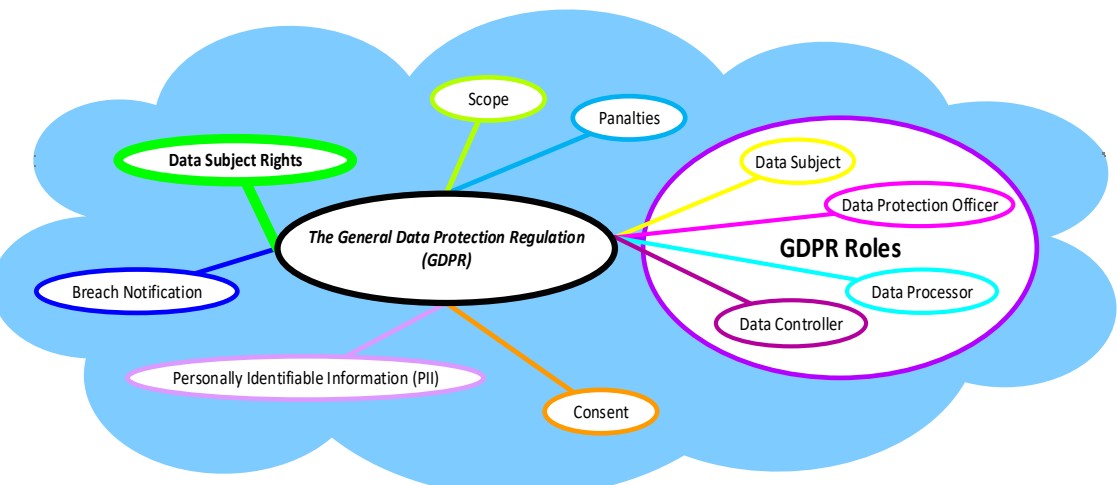

**Figure 1.** Overview of the key elements of the General Data Protection Regulation (GDPR).

GDPR is applicable globally to any organisation that collects, stores, processes or monitors a European residents' personal data, regardless of location and nationality, and includes free goods and services [6]. It covers both computerised and hard copy fileable data. Under GDPR, the data protection authorities will have the ability to impose sanctions with possible publicity and can impose significant fines of up to EUR 20 million [1]. Compensation may also be payable to individuals whose rights have been breached. GDPR introduces four new roles, which are:

- Data Subject: A data subject is a natural living person who can be identified directly or indirectly [1]. A data subject is anybody residing in the EU, not just EU citizens.
- Data Protection Officer: A Data Protection Officer (DPO) (Data Protection must have specialist skills and expertise to oversee GDPR compliance, ensuring obligations are met from the highest level of management; they are the point of contact for the supervisory authority and monitor the organisation's compliance with the law. The DPO can either be an employee or outsourced service. Under GDPR, whilst it is mandatory for all organisations to appoint (DPO), small organisations with less than 250 employees are exempt.
- Data Processor: A data processor is an organisation that process data as per instructed by their data controller like cloud hosting providers [5]. GDPR recognising the complexity of modern-day data processing relationships identifies that data processors play a vital part in the protection of European citizens data and so introduced direct rules for data processors such as record keeping and reporting data breaches.

- Data Controller: The data controller is the organisation that collects, processes and stores PII and must be able to demonstrate GDPR compliance, which means the burden of proof lies with them [7].

Under GDPR, consent must be freely given, specific, informed and unambiguous [8]. All EU contracts must be valid and reflect the individual's new rights. PII is any information that can be used on its own, or combined with another bit that can be used to identify a living EU resident such as name, address, IP address, Personal Public Service Number (PPSN), account details, etc. PII is either Sensitive PII or Non-sensitive, where sensitive PII could cause harm to a data subject if breached, and therefore, must be encrypted both in transit and at rest [9]. Whilst non-sensitive PII will not cause harm to a data subject, and therefore can be unencrypted. The key obligations imposed on organisations by GDPR are illustrated.

- Data Protection Impact Assessment (DPIA): DPIA aims to identify potential risks involved the collection, processing and storage of PII, the impact on the privacy of the data subject and identify ways to mitigate those issues [8].
- Transparency: An organisation must have a granular level of transparency into their PII from consent, collection, processing and storage for the full life cycle of that data and mandatory clauses (EU, 2016).
- Data Minimization: PII can only be collected and processed where there is an identifiable reason why it is needed and should be kept no longer than is necessary for the purpose for which it was collected, and no additional data can be obtained [8].
- Security: Organisations must ensure that technological and organisational methods are in place to securely protect PII as per industry standard and best practices [7]. Implementing an IS0 27001 compliant ISMS would assist in achieving compliance [10].

A data controller must report a data breach to the data Protection Commissioner within 72 h and notify data subjects unless there is no risk of harm [7]. GDPR has increased and strengthened the rights of a data subject [11].

*Right to Erasure ("Right to Be Forgotten")*

Article 17 of the EU General Data Protection Regulation (GDPR), the Right to Erasure ("Right to Be Forgotten"), was originally known as Right to be Forgotten (RTBF) but is now called the Right to Erasure [12]. This right proves to be the toughest data subject right to operationalise and even the second most difficult GDPR obligation in practice overall [13]. The Right to Erasure ("Right to Be Forgotten") is a fundamental data subject right to ask from a controller that all their PII be erased and the controller must do so without undue delay and free of charge in accordance with GDPR Article 17 [8]. This right does not only apply to search engines, but to any organisation that collects, processes or stores PII. If you used to be an Eircom customer and you are not anymore, then you can ask them to remove it. That is the Right to Erasure ("Right to Be Forgotten". The term "Right to be Forgotten" is a concept which originated from individuals need to "determine the development of their life in an autonomous way, without being perpetually or periodically stigmatised as a consequence of a specific action performed in the past." [14]. This concept has been practiced in the European Union (EU) and Argentina since 2006 [15] and there have been many discussions and debates over the years surrounding it with regard to its vagueness and concerns about its impact on the right to freedom of expression, its interaction with the right to privacy and whether creating a RTBF would decrease the quality of the internet through censorship and re-writing of history. Other concerns relate to problems such as revenge porn sites appearing in search engine listings for an individual's name or references to petty crimes committed many years prior still linked and displayed as part of an individual's footprint [16]. In 1995, the EU adopted the European Data Protection Directive, Directive 95/46/ec, to regulate the processing of personal data aiming to secure potentially harmful private information relating to an individual [16]. On 13 May 2014, in the Google Spain v AEPD and Mario Costeja González case, the European Court of Justice ruled that people have the right to be forgotten, solidifying it as a human right.

The irony of it all is that Mr Gonzalez intention was to obscure that information, but it resulted in becoming worldwide publicity. Courts worldwide have been referring to the European Court of Justice (2014) ruling on the right to be forgotten [17]. Then, in 2016, under the introduction of the General Data Protection Regulation, this principle was modernised to bring it in alignment with digitalisation [18]. Grounds upon which a data subject can exercise the right to be forgotten are as follows [18].

- The data is no longer required for the purpose that it was originally collected.
- The data subject withdraws consent.
- The data subject objects to the processing and there are no overriding legitimate grounds.
- The PII was processed unlawfully.
- The PII must be erased for legal obligations.
- Processing of children's PII collected via information society services.

Organisations must erase PII upon receipt of a valid request and this must be done within 30 days and free of charge [19]. If it is not carried out and without undue delay, then the data subject can report this to the Data Protection Commission. So, organisations now need to be concerned about their employees, customers and suppliers as well as authorities. Organisations must also erase PII once it expires. This is quite a complex task for most small organisations and they must understand what PII they retain, why they need this data, how long it can be retained, and they need to identify and locate PII throughout the entire hybrid cloud infrastructure including excel, work, PowerPoint, backups, etc. Most do not have a clear understanding of where all the PII they retain is stored, including third parties. In fact, with the expanded definition of PII, they may not have a full understanding of all the data that should be classified as PII. Adding to this is the complexity of the data landscape within a hybrid cloud infrastructure. Additionally, they will not have the expertise or resources needed to undertake such a task. When using cloud services or third parties, both must understand what PII they have and why they have it and the liabilities involved. Retaining expired PII is a liability because if a breach occurs, compensation pay outs will not only apply to existing clients but also to clients an organisation no longer has. Organisations must have a system in place to easily identify, locate and report all PII for that one data subject and a system to identify, locate and report all PII that has expired, so it can be reviewed and deleted promptly. They must have this documented, so it can easily be followed and used to demonstrate that they have procedures in place to meet compliance. Automation will be an important part of this compliance to identify, locate and report all PII as having employees randomly looking through personal data would be a privacy issue. Many organisations tended to store extra data in case it may be useful later as storage was cheap, and it was easier than putting processes in place to check for obsolete data and removing same, now this must be erased and only the data relevant retained.

It may be impossible to truly enforce the right to be forgotten, e.g., data is really outside the control of an organisation with the use of smart phones which enables an individual to take pictures of personal data, or an individual taking a screen print etc., and these could be distributed to various other locations by the click of a button using their private email, or removable devices. Another consideration is deleted files are not erased as they are still contained on the hard drive, even after emptying the recycle bin, thus enabling the recovery of PII [20]. It can be impossible to delete a single record for some PII without impacting on other PII, e.g., microfiche; therefore, it is not feasible to destroy this without losing other data that is still required by the organisation [21]. There are also built in features like Volume Service Shadow (VSS) whereby data can easily be recovered once deleted [9]. Deleted data can also be recovered in an SQL server database using Log Sequence Numbers (LSNs) or by using a third-party software like SQL Database Repair [22]. Data deleted is recoverable but if erased properly is permanently removed [23]. In some cases [24], it is possible to recover almost all deleted browsing activity. PII can be held on any device that has permanent memory like desktop, printer, laptop, external hard drives, etc., so deciding whether to overwrite or destroy will depend on whether the organisation will use the

device again [10]. With the introduction of GDPR, small organisations must monitor and manage their PII. Under GDPR, PII references any information that can be used to identify a specific living individual. Personal identifiers are displayed in the diagram above; however, due to technology, the scope has expanded to include IP address, login credentials, social media posts, geolocation, biometric, genetic and behavioural data. This expanded scope increases security and privacy challenges. Adding to this mixture is the challenges of direct and indirect personal data/information [25]. GDPR is applicable to automated PII, manual filing and pseudonymised PII [10]. Under GDPR, personal data references special categories of personal data [8], which include genetic data and biometric data that uniquely identifies an individual. Exclusions are data relating to crime [26]. Personal data can be broadly categorised as structured, semi-structured and unstructured.

Structured data/information references data that is highly organised, for example, data stored in a relational database like SQL or stored in an excel spreadsheet. This type of data is easy to find, filter and search [27].

Semi-Structured-Data/information references data that cannot neatly fit inside a relational database; however, it does have some structural properties allowing for analysis [28].

Unstructured data/information references data which is unorganised and does not have a pre-defined model. It cannot neatly fit inside a relational database and is incredibly difficult to identify, locate, manage and use, like word. This data does not fit into relational databases and is the data that organisations struggle with when trying to meet Right to Erasure ("Right to be Forgotten") compliance, as it is impossible to scrutinise, and therefore must be metamorphosed into structured format, otherwise it is of no use to the organisation [27]. Unstructured content is typically text-heavy and multimedia, which is estimated to represent more than 80% of the overall business information created and used. The volume of unstructured data held in various repositories within a hybrid environment increases continuously, resulting in the identification and location of same becoming more and more difficult to manage [26].

## 3. Cloud Computing

Cloud computing hosts and delivers various services over the Internet to store, manage and process data [29]. It has had a remarkable effect on Information Technology as cloud providers like Google, Amazon and Microsoft compete to make their cloud platforms the most powerful, cost effective and reliable. This in turn enables organisations to improve their business models, and they no longer must plan for provisioning as resources are allocated according to the level of demand. One important aspect of the cloud is that cost is normally in proportion to demand, which can be influenced by performance requirements. Resources must be allocated efficiently to ensure effective planning of costs and resources for both the client and the service provider [29]. Cloud service providers aim to offer methods to allocate or deallocate resources on demand to meet the service levels in the contract, or Service Level Agreement (SLA). Cloud computing has four deployment models [27]. A deployment model defines the purpose of the cloud and the nature of how the cloud is located. "The NIST Definition of Cloud Computing" classified cloud computing into four cloud types (public, private, community, and hybrid), and also classified cloud computing into the three SPI service models—SaaS, IaaS, and PaaS [29]. In Infrastructure as a Service (IaaS), clients can provision virtual machines, virtual storage, virtual infrastructure, etc. The service provider is responsible for the management of all the infrastructure, whilst the client is responsible for all the other aspects of deployment including operating system, applications, user access. In Platform as a Service (PaaS), clients can provision virtual machines, operating systems, applications, services, deployment frameworks, transactions and control structures. Clients can also deploy their own applications on the cloud infrastructure or use applications and tools supported by the service provider. The service provider is responsible for the management of the cloud infrastructure, the operating systems, and the enabling software, whilst the client is responsible for installation and management of the application they deployed. Software as a Service (SaaS) is a complete

operating environment with applications, management, and the user interface. An application is provided to the client through a thin client interface (a browser, usually). The service provider is responsible for everything, from the application down to the infrastructure, whilst the client's responsibility starts and ends with entering and managing its data and user interaction. SaaS is on demand software which is charged on a pay per use basis.

*Hybrid Clouds*

A hybrid cloud is a cloud computing infrastructure integrating multiple different cloud models (public, private or community), each retaining their unique characteristics, but are bound together as one unit. It offers standardised or proprietary access to data and applications and application portability. This concept also entitled as cloud bursting according to [26]. With this model, an organization utilises their own computing infrastructure to handle their normal requirements, but any spike in requirements that occurs will be handled by public cloud services. There are many issues like cloud inter-operability and standardization in hybrid cloud computing model. Critical activities can be performed within the private cloud and the non-critical activities performed within public cloud, according to [29]. Advantages include scalability of an on-demand, externally provisioned cloud whilst also availing of increased security, privacy and auditability. It provides a variety of options that can be utilised via public or private clouds, whereby an organisation can select the most cost-effective delivery method for agile business requirements whilst staying within strict security and service level agreements. Disadvantages are that applications are spread across different environments adding complexity and the need to increase management and monitoring within the environment. It is best suited to organisations that need support for non-critical applications, great scalability, flexibility and optimal service levels together with the need for new agile environments requiring new services to be available immediately. The hybrid is ideal for an organisation utilizing private cloud that incurs peaks in demand requiring resource elasticity, but the cost of permanently having the benefit of resource elasticity far outweighs the access costs of on-demand. A hybrid cloud enables the small organisation to take advantage of the benefits of public cloud, where they can keep PII in a private cloud, giving them the option of moving to the cloud gradually, if they so choose. A key issue with hybrid cloud is that many organisations have haphazardly moved into it rather than having chosen a hybrid strategy.

Many of the key benefits of hybrid is that with public cloud you obtain hardware, networking, storage, service and interfaces owned and operated by a third party for use by other organisations or individuals. Whilst there are a variety of public cloud service providers available, Amazon Web Services (AWS) was selected for this test scenario as it provides Free Tier, it practices ISO 27k industry standards and PCI DSS best practices, and is the most popular. Having the private cloud, whilst like public cloud, in that you obtain hardware, networking storage, service and interfaces, is however, ultimately owned and operated by the organisation. This private cloud will be secured to the on-premise environment. Many organisations chose hybrid as the total cost of ownership (TCO) with cloud solutions is far lower than ongoing costs of maintaining on-premise hardware. However, there are organisations already using public cloud yet are looking to private cloud solutions to reduce costs, particularly with high volumes of data as this requires more storage and network charges. So, it is about finding the right mix of public and private cloud solutions to gain cost savings. The hybrid cloud allows organisations to build redundancy into their IT architecture giving them extra security in the event of Disaster Recovery (DR). It also provides scalability if they need to scale up or down depending on spikes and troughs. Databases in the hybrid cloud are handy for new applications when you are not sure how successful they will be in the marketplace. Many organisations may want to sell the application fast and cheap; therefore, using public cloud resources for new untested applications before through the capital expenditures associate with launching in a private cloud. A hybrid cloud is beneficial for cloud bursting, so workloads can spill over to another cloud to meet capacity demands. Providing a high available geo-redundant

setup using private cloud can be expensive to build. Many organisations cannot justify such expenses yet without one the organisation is vulnerable.

A hybrid cloud infrastructure comprises of two or more different cloud infrastructures, private, community, and/or public, that remain exclusive entities, but are bound together by standardised or proprietary technology that enables data and application portability (e.g., cloud bursting for load balancing between clouds) [30]. The ISO/27k family of standards aim to help organizations, regardless of size, secure their information. These standards provide requirements for an information security management system (ISMS). The ISMS is management framework enables organisations to identify, analyse and address information risks. It ensures security arrangements are perfected to keep pace with the ever-changing security threats, vulnerabilities and business impacts which is crucial part in such a dynamic field. ISO27k's flexible risk-driven approach is advantageous compared to PCI-DSS.

## 4. Hybrid Cloud Test-Bed Design

We outline a test scenario for a small organisation that stores PII in a variety of data formats in various locations throughout a hybrid cloud environment. This test scenario is to examine if the identification, location and reporting of PII for specific conditions can be successfully carried out within a hybrid cloud environment, and what challenges it presents. This research aims to produce guidelines for a small organisation with a hybrid cloud environment, so they can become compliant with Article 17 of GDPR, right to be forgotten principle. We outline the setting up of a hybrid cloud test scenario. ISO/IEC 27001:2015, ISO/IEC 27002:2015 and ISO/IEC 27018:2014 were consulted when setting up the test environment and the PII. ISO/IEC 17788:2104 and ISO/IEC 17789:2014 and ISO/IEC 19944:2017 were consulted when setting up the cloud components. Implementing a hybrid cloud environment is really setting up three different environments and then ensuring they are integrated, all the while enforcing data privacy. PII for a data subject was created in various data formats in various locations. Once the environment was set up, scripts were created to securely identify, locate and retrieve the location of the relevant PII all the while adhering to data privacy. The ad hoc hybrid environment consists of a Local Area Network (LAN) containing the local office, a Wide Area Network (WAN) containing the on-premise private cloud infrastructure, and a public cloud infrastructure (see Figure 2).

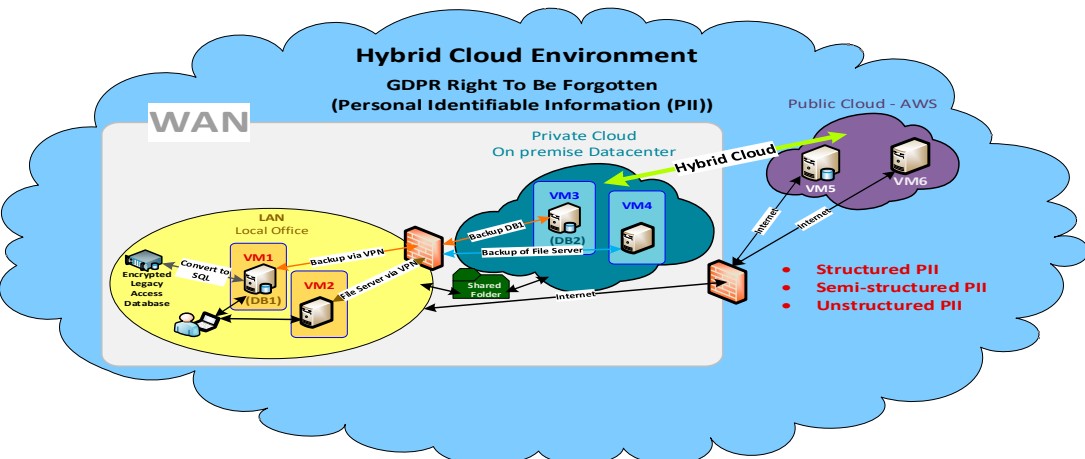

**Figure 2.** Overview of the Test Hybrid Cloud Environment.

Within the LAN, the local office has a physical server and a virtual server. The physical server holds an encrypted legacy access database storing personal/sensitive data. The Virtual Machine named VM1, contains the main production SQL database server named DB1 (which is the "SampleDatabase"). As the business progressed, the original production

access database was upgraded to the SQL Server database DB1. The management laptop was a Windows 10 64-bit with VMware workstation 14 pro. Virtual machines consist of a gold image virtual machine and a virtual machine containing an SQL database. The WAN encompasses the on-premise private cloud infrastructure and the public cloud infrastructure. The private cloud is accessed via a FortiClient IPsec VPN, whilst the public cloud is accessed via the internet. The private cloud environment consists of an ESXi server—windows containing two virtual machines (VMs) named VM3 and VM4, one of which will be used to store the Production SQL database whilst the other will be used to store critical documentation. VM3 contains an SQL database server named DB2, utilised as a backup of the production SQL database DB1. This caters for redundancy and provides high availability. VM4 contains a is a backup of VM2, catering for load balancing, redundancy and high availability. A shared folder will be used between the local office and the private cloud. The management laptop can access all VMs. All databases on the WAN retain critical personal and sensitive data. The public cloud infrastructure is accessed via a Virtual Private Cloud (VPC) and is utilised to store backups. It contains a further two VMs named VM5 and VM6. VM5 is an RDS instance used as a backup of the Production SQL database VM3. VM6 is an EC2 instance to hold a backup of the various types of archived documentation like reports, excel spreadsheets, word documents, PDFs and CSVs.

*4.1. LAN, Private and Public Cloud Setup*

This experiment will focus on performing data discovery on a collection of structured, semi-structured and unstructured data, to identify and locate all PII within this hybrid environment for (1) PII relating to a specific data subject and (2) PII relating to a specific date. On both occasions, a report will be produced containing the location of the PII which will be reviewed and if valid deleted without undue delay. An inventory of all the hosts detailing host name and IP address contained on the network was carried out and the PII was identified and classified. To carry out this data discovery, a Super User is setup with AAA. This user must have authentication (valid username and password), and where possible a two-factor authentication, authorisation to carry out the relevant activities, issue commands and to run PowerShell scripts and accounting to gain access and control to all the relevant devices, databases, folders, files and documents in all sections of the organisation, both internal and external. The Super User must have the correct enforcement policies set in place to be able to get to the data they need and have permission to use this data.

The hybrid cloud is set up to contain three components, a LAN, a private cloud and a public cloud. This small organisation will have one person allocated to the task of Right to Erasure ("Right to be Forgotten"). As a hybrid cloud involves three separate environments, the user will be assigned administrative access to everything in each of the components. This user will be assigned a secure dedicated management laptop. It will be set up to enable connectivity with every device within the hybrid cloud infrastructure. The rest of this chapter details the setting up of each component of the hybrid environment.

4.1.1. Local Area Network (LAN)

As illustrated in Figure 3, the LAN consists of a management laptop, a virtual machine and a further laptop acting as a server to store the legacy encrypted access database, which is locked away in a secure cabinet.

VMware Workstation 14 Pro for Windows 10 64-bit was installed onto the management laptop and the relevant licences applied to enable the creation of virtual machines. A trusted SSL certificate was downloaded via vSphere and FortiClient were installed on also to enable VPN connection with private cloud. The relevant modules for AWS and SQL were imported into PowerShell. Two virtual machines were created in VMware workstation. A shared folder was created between the management laptop and VM1. An Access database was created and named *SampleDatabase.accdb*. This database was upgraded to the SQL database stored on VM1 within the VMware Workstation.

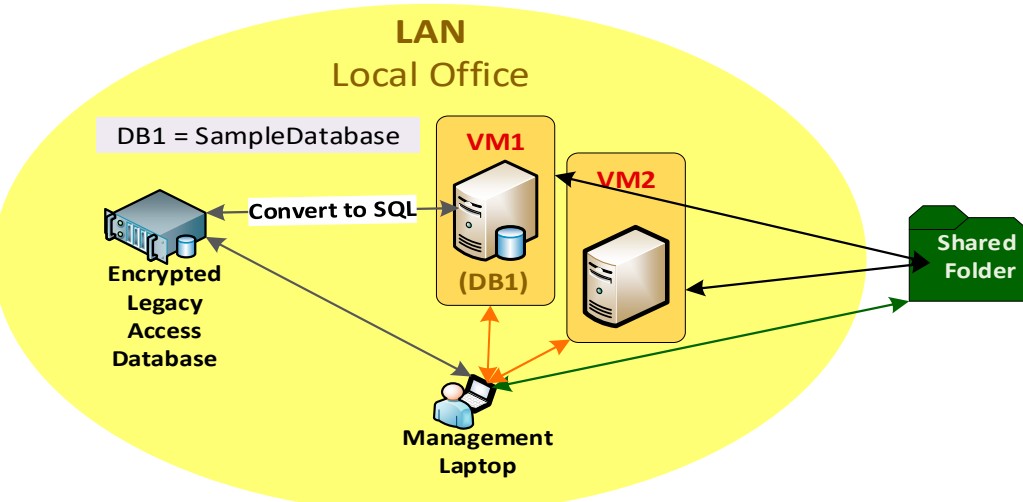

**Figure 3.** Implementation of Local Area Network (LAN).

4.1.2. Private Cloud Environment

Figure 4 shows the private cloud environment consisting of two VMs. A physical server ESXi-6.0 in the LYIT Computing Data Centre (CDC) was configured with an up-to-date ESXi licence key, an IP address, default gateway, DNS Server and hostname of ESXi-15. A vSphere standard switch was created and a VLAN with ID 139 was added to the switch. To enable secure access with encryption, the VMware vSphere Client (with a trusted SSL Certificate) was downloaded and installed onto the management and FortiClient 5.6 for windows was downloaded, to install secure remote access, and IPsec VPN configured. VM1 will contain structured data. A further VM called VM2 was created and configured to act as a file server that stores documents. It was populated with a variety of semi-structured and unstructured PII in various stored in various locations within its file structure.

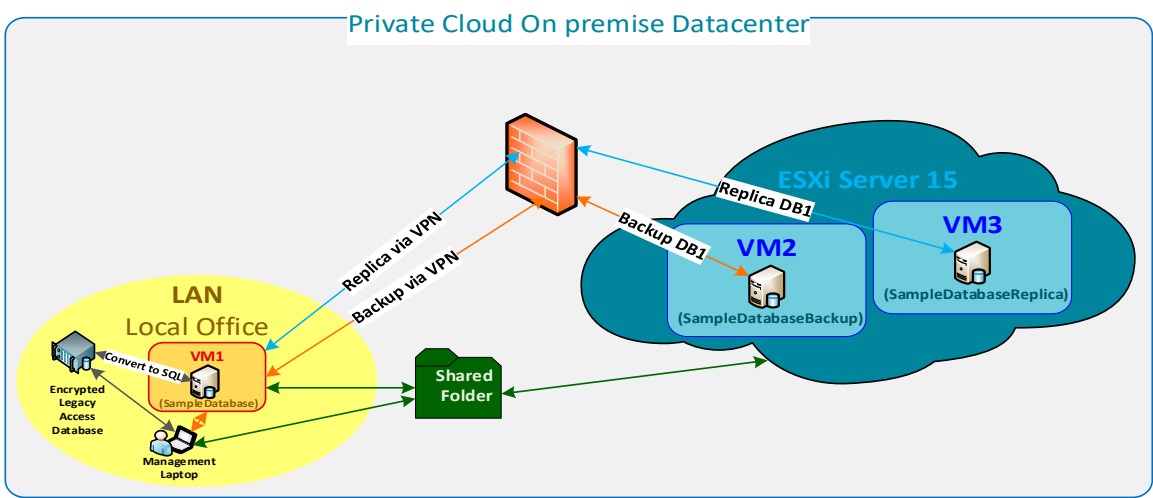

**Figure 4.** Implementation of Private Cloud Environment.

4.1.3. Public Cloud Environment

Amazon AWS was the selected cloud provider as it has a large customer base that has gone through rigorous security, is ISO 27k certified and provides Free Tier. Whilst it provides Free Tier, there is no mechanism in place to stop you from going over the Free Tier limit. An EC2 instance was created to act as a file server which will hold a variety of documents types containing PII for a data subject and PII with an expiration date. An RDS

instance was created as backup to the SQL production database. A Virtual Private Cloud (VPC) is created by default, which can be configured to suit the organisation. A security group was set up for each class of instance. Adding and restricting ports can be a bit messy, but for this test scenario, ports 3306 must be set up for databases and port 80 for http. It is best practice to setup granular security groups instead of general ones.

### 4.2. Data Discovery

PII stored within a hybrid cloud infrastructure can be in various formats and locations. Therefore, a small organisation would first need to identify, locate and document all the hosts in their network. There are many free tools available to download that can be run to discover the hosts on the network, e.g., Kali Linux. Then, they can identify what data is retained that would be considered PII, and ensure this data is secured and protected. The first area for review should be the high-risk areas that involve financial data or third parties. A review of all the business processes must be carried out to ensure PII was captured with valid consent and has a valid need for processing as well as how long the data is to be processed. As PII can be either direct or indirect, this will focus on the direct identification of the data subject as the research is about identifying and retrieving the location of PII for a data subject or a specific date. As this PII can be stored in a variety of data formats, it was decided to use different methods for identification as follows:

(a)   Structured

From the database schemas the following fields listed in Table 1 were classified as direct PII.

**Table 1.** Classification of Direct Personally Identifiable Information (PII).

| Employee | Customer | Payment |
|:---:|:---:|:---:|
| employeeId | customerId | PaymentId |
| empLastName, empMiddleName & empFirstName | custLastName, custMiddleName & custFirstName | cardNumber, sortCode & accountNumber |
| empAddressId | custAddressId | |
| empContactNumber | custContactNumber | |
| empEmail | custEmail | |

To search for PII for a data subject, a query was created to search the database for a data subject using "empLastName" and "empFirstName". The search variables can be changed to "empEmail" or "empContactNumber" or a combination of them all. To search for PII where the contractual date has expired, a query was created to search the database and report records where the "lastaccessdate" is equal to a date variable.

(b) Semi-structured

A PowerShell script was created to search through all the folders in each drive for documents of type .csv and .xml. To search for PII for a data subject this script will search the content of each document type .csv and .xml for variable "LastName" and "FirstName". Other variables can be added to be more specific like Date of Birth, PostCode. To search for PII where the contractual date has expired, the script will search through all folders in each drive for date created/modified is equal to a date variable.

(c) Unstructured

A PowerShell script was created to search through all the folders in each drive for documents of type .pdf, .docx, .txt and .xlxs. To search for PII for a data subject this script will search the content of each document type .pdf, .docx, .txt and .xlxs for variable "LastName" and "FirstName". Other variables can be added to be more specific like Date of Birth, PostCode. To search for PII where the contractual date has expired, the script will search through all folders in each drive for date created/modified is equal to a date variable.

## 5. Testing

We detail possible PII breaches an existing small organisation might have prior to implementation of our recommendations. We look at the measures taken to check the possibility of finding documentation containing PII for a data subject and PII for a specific date throughout a Hybrid Cloud Environment and how these PII breaches have been mitigated. The focus is on the direct identification of the data subject as the research is about retrieving the PII for a data subject or a specific date. We examine the challenges a small organisation using a hybrid cloud may face in becoming "Right to Erasure" compliant to demonstrate that it is possible to be compliant, given a set of recommendations.

### 5.1. Example Scenario

Upon a valid request from a data subject, an authorised user would be assigned to create and run a query over the production database, to identify all records relating to the data subject. The results would be reviewed, and confirmation given to proceed with deletion of the selected records. Figure 5 highlights the potential PII breach of GDPR Article 17 "Right to Erasure". It illustrates that structured, semi-structured and unstructured PII is held on every device within each component of the hybrid cloud, not just the production database. It also highlights that out of the multiple users within the hybrid environment, users 1, 2, 5 and 6 could be assigned this task.

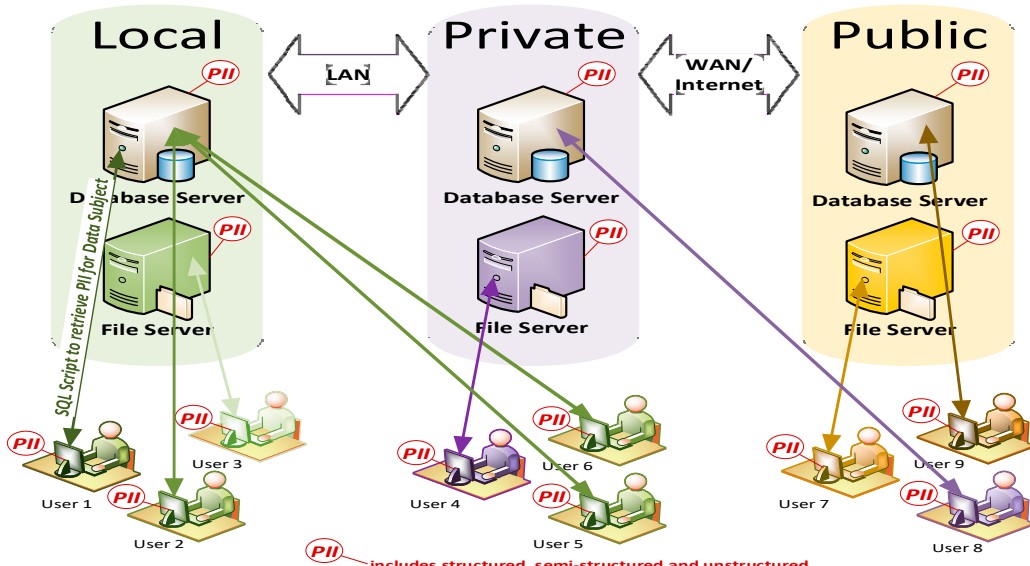

**Figure 5.** PII Location—A Potential Breach of GDPR Article 17 "Right to Erasure".

Many organisations would be in breach of Article 17 as they:

1.  Do not have secure remote connectivity set up to access all devices on each component within the hybrid cloud; therefore, all PII cannot be identified or accessed on every device. In this example scenario, the assigned user only has access to the production database.
2.  An SQL query would be run to identify and report the location of PII in the production database, without using encryption.
3.  The output from the query would possibly be stored in a document/file that was not encrypted.
4.  Do not have audit trails in place that can be used to demonstrate reasonable measures were taken to identify, locate, report and delete PII. This example highlights there is no guarantee which user will be assigned these tasks thus making auditing and event logging harder to trace.
5.  Do not have authority to access all devices and the devices they have access to, they do not have authority to all PII so unable to identify, locate and report PII.

6. User does not have access to some of the passwords or cryptographic keys, and therefore cannot access all PII.
7. Retains expired PII.
8. Retains more PII than what was/is required for the purpose, thinking they might use it in the future.
9. Do not know their data landscape, nor what constitutes PII and as a result upon receipt of a valid request from a data subject to erase their PII, think deleting PII from the production database will suffice. In this example, the assigned user only checks the production database.
10. PII stored in other formats were not investigated.
11. An SQL query would have been run only over the Production database to locate PII, so other databases and data formats throughout the hybrid cloud environment would have been overlooked.
12. Expired PII may be deactivated in some way, but unlikely to have been identified with a view to erasure.
13. No automation tool or script to identify and report the location of PII for a data subject or PII that has expired.
14. No access control lists or firewall rules configured to enable a user or device to access PII on every device.
15. The backup process of nightly, weekly and monthly would erase the PII; however, there could be occasions where the monthly runs late, so PII would not be erased within the time limit of 30 day, which would be a PII breach.
16. Backups and archives may be stored off-site and on tapes.
17. PII would only have been deleted from the production database.
18. No processes or procedures in place to document.

### 5.2. Tests

Scripts using encrypted username and passwords were created on the management laptop to search for a data subject, to search for a specific date. These scripts will report the location for the specified criteria. Then, using PowerShell ISE remoting, these scripts will be copied to each device and executed. An overview of the test structure is displayed in Figure 6.

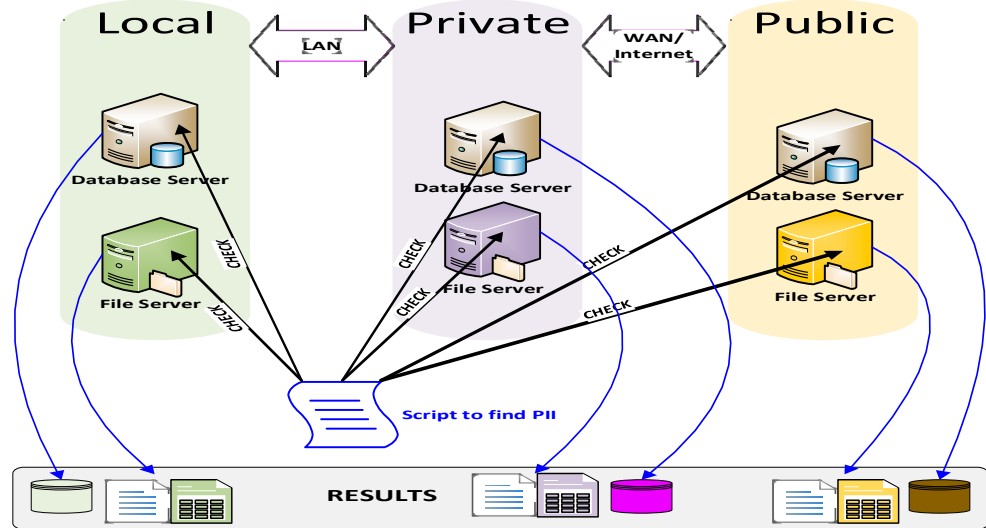

**Figure 6.** Overview of the test structure.

The PII will be found in various locations within the hybrid cloud environment and will be stored as a combination of (a) Structured: Access database and SQL database; (b) Semi-structured: .csv and .xml; (c) Unstructured: .docx, .pdf, .txt and .xlxs and (d)

Encrypted: .zip, Legacy Access Database. A small sample of test data was set up for a data subject named Philomena Ann Kelly, and a date was entered for the contractual expiry date. PowerShell scripts were created to retrieve PII for a data subject and PII for a specific date. The aim is to investigate if the relevant data formats can be found on the various hosts within the hybrid cloud and interrogated to retrieve PII for a data subject (named Philomena Ann Kelly) and PII to expire. Each component of the hybrid environment was configured to enable the execution of the PowerShell scripts to connect, identify and retrieve the location of both PII for a data subject and expired PII.

### 5.2.1. Structured PII Held in Databases within the Hybrid Cloud

The LAN "SampleDatabase" is a SQL database which is stored on a virtual machine VM1 within VMware Workstation 14 Pro. It was accessed via SSMS using windows credentials. PowerShell commands were run as Administrator on the management device. The script invoked a query to search the database for a data subject where "empLastName" and "empFirstName" is equal to the data subject name supplied. PII was successfully retrieved for a data subject named Philomena Ann Kelly, as shown in Figure 7. The private cloud "SampleDatabase" is an SQL database which is stored on a virtual machine, VM3, on the ESXi-15 server. PII was successfully retrieved for the data subject named Philomena Ann Kelly from the private cloud database, as shown in Figure 8. The public cloud "SampleDatabaseBK" is an SQL database which is stored on Amazon AWS RDS—public cloud. A connection was made from the management device using SSMS on VM3 connect to the AWS RDS using the RDS endpoint and SQL server authentication. PII was retrieved successfully for data subject Philomena Ann Kelly from the AWS RDS database.

```
SQLQuery1.sql - VM...Administrator (53))*  ⊕ ✕
USE SampleDatabase
SELECT * FROM tblCustomer
WHERE custLastName = 'KELLY' AND custFirstName = 'Philomena'
```

| | customerId | custLastName | custMiddleName | custFirstName | custAddressId | custAddressType | custContactNumber | custEmail |
|---|---|---|---|---|---|---|---|---|
| 1 | 1012 | Kelly | Ann | Philomena | 500017 | R | 0166234324 | pak@aol.com |

**Figure 7.** PII retrieved from LAN database for data subject.

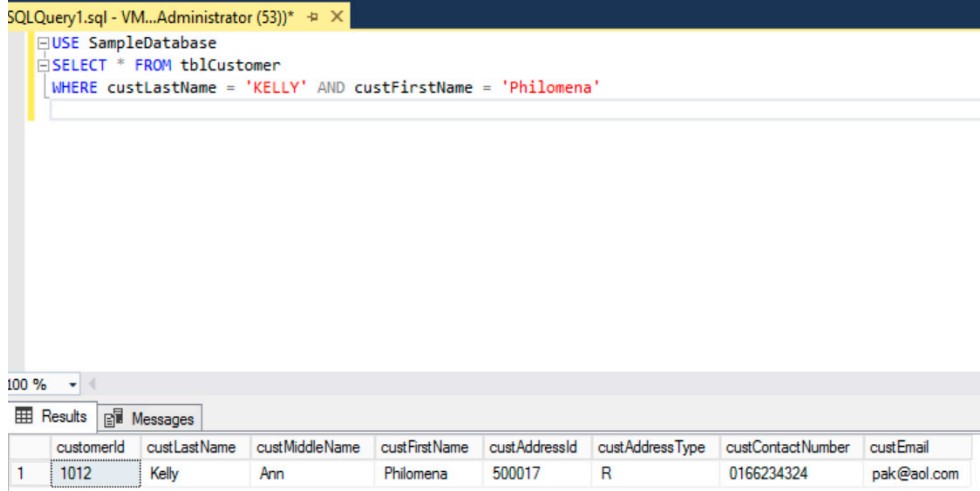

**Figure 8.** PII retrieved from private cloud database for data subject.

### 5.2.2. Semi-Structured PII Held in Various Locations within the Hybrid Cloud

We ran a PowerShell script to browse through the LAN folders and sub-folders in each drive and retrieves the directory path for all documents of type .csv and .xml and writes the directory path to a file. It then read through each of these paths and searched the content of each document type .csv and .xml for variable that matches the data subject name. We retrieved PII for data subject named Philomena Ann Kelly stored in the documents shown in Figure 9. We ran a similar script on the private cloud and retrieved PII for data subject named Philomena Ann Kelly stored in the documents shown in Figure 10. The same was done on the public cloud and resulted in the documents shown in Figure 11.

doctest4loc.csv

log4netconfigtest5loc.xml

**Figure 9.** LAN documents found.

doctest4Priv.csv

log4netconfigtest5Priv.xml

**Figure 10.** Private cloud documents found.

doctest4pub.csv

log4netconfigtest5pub.xml

**Figure 11.** Public cloud documents found.

### 5.2.3. Unstructured PII Held in Various Locations within Hybrid Cloud

We ran a PowerShell script to read through all the folders and sub-folders in each drive, retrieve the directory path for all documents of types .pdf, .docx, .txt and .xlxs, and write the directory path to a file when found. It then reads through each of these paths and searches the content of each document type .pdf, .docx, .txt and .xlxs for PII that matches the data subject name. We retrieved PII for subject Philomena Ann Kelly stored in the documents shown in Figure 12. Figure 13 shows the retrieval of PII from the private cloud and Figure 14 shows the results from the public cloud search.

TempDoc DatabasesAndLocationtest6loc.docx

PKiesebergtest7loc.pdf

list_sorted_by_LastWriteTime_Descendingtest8loc.txt

Test9loc.xlsx

**Figure 12.** LAN documents.

PKiesebergtest7Privh.docx

PKiesebergtest7Privh.pdf

list_sorted_by_LastWriteTime_Descendingtest8Priv.txt

Test9Priv.xlsx

**Figure 13.** Private cloud documents.

TempDoc DatabasesAndLocationtest6pub.docx

PKiesebergtest7pub.pdf

list_sorted_by_LastWriteTime_Descendingtest8pub.txt

Test9pub.xlsx

**Figure 14.** Public cloud documents.

*5.3. Post Implementation*

By implementing the recommendations for "Right to Erasure", upon a valid request from a data subject, the super user would be assigned to securely identify, locate and retrieve the location of the relevant PII. The super user will manually log on to each device using the encrypted credentials, then execute the relevant script on each device to identify, locate and retrieve the location of the relevant PII.

Figure 15 demonstrates that PowerShell can securely access every device within the hybrid cloud environment. It can identify, locate and report the location of PII for both a data subject and expired PII on every device on each component throughout the hybrid cloud environment.

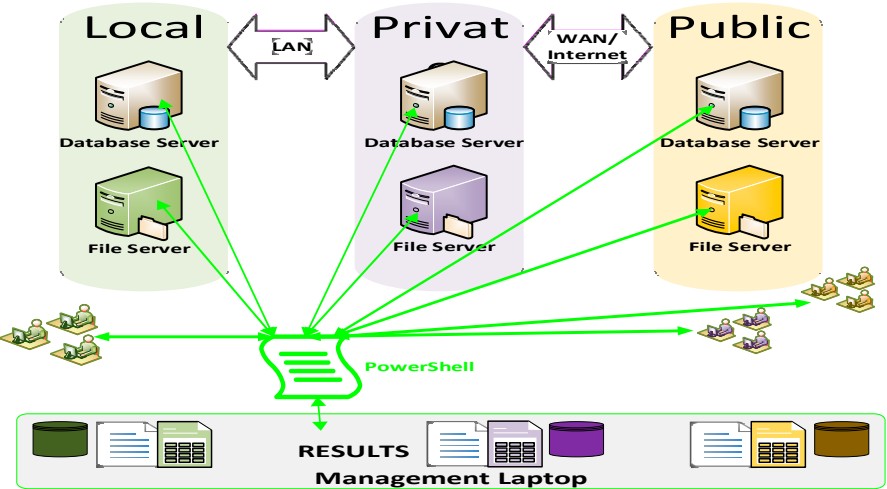

**Figure 15.** PII Location Breach of GDPR Article 17 "Right to Erasure" Eliminated.

This will eliminate the PII breaches as:

1. The dedicated management laptop has a file containing all the relevant credentials encrypted. These can only be decrypted by the super user (who encrypted them) and only on this device (where they were encrypted), enabling secure non-intrusive remote connectivity to every device on each component within the hybrid cloud, enabling all PII to be identified and located.
2. PowerShell can encrypt and decrypt so files containing PII will be encrypted in transit as well as at rest.
3. Credentials are stored encrypted and only the super user can decrypt these when required.
4. Files can be encrypted using an Advanced Encryption Standard (AES) algorithm, so the data retrieved using the script may be encrypted. The use of specified trusted hosts adds to securing access.
5. Only one super user will carry out this task; therefore, audit trails and event logs can be used to check the activity of this user and demonstrate that reasonable measures were taken to identify, locate, report and delete PII.

6. The super user has the appropriate authority to access all devices and access all PII on these devices so PII can be identified and the location reported for all PII within the hybrid cloud environment.
7. The super user will have access to all passwords, cryptographic keys, names and IP addresses of all devices within the hybrid cloud enabling administrative access to all devices.
8. Expired PII is no longer retained as PowerShell can identify and retrieve the location of all PII within the hybrid environment and so can be deleted.
9. Whilst identifying all the PII that is currently retained, extra PII may be identified and dealt with, but data minimisation would be carried out under GDPR Article 5 "Principles relating to processing of personal data".
10. The data landscape is clear so now all areas containing all formats of PII can be identified and located within the hybrid cloud environment.
11. The super user has administrative authority to all PII on every device within the hybrid environment and because of this, the process cannot be fully automated. A manual log on is required to every device and each script run separately. All PII can now be identified for a data subject and PII where the contract has expired.
12. With all devices configured, PowerShell can be run to locate and retrieve the location of PII for both a data subject and expired PII.
13. Access control lists and firewall rules are configured to only allow the dedicated management machine and super user access to PII on every device.
14. The backup process of nightly, weekly and monthly would erase the PII; therefore, procedures are put in place to ensure that the monthly is run on 28th day or nearest weekend to 28th day of the month.
15. Backups and archives may be stored off-site and on tapes. These must all be encrypted and stored in a secure environment and where feasible erased.
16. All PII would be securely deleted from every device within the hybrid cloud.
17. Steps undertaken would be documented so they can be used to demonstrate that reasonable measures were taken to be compliant with GDPR Article 17, "Right to Erasure".

Table 2 summarises the GDPR Article 17 "Right to Erasure" breaches prior to the implementation of the recommendation and how these have been eliminated post implementation, thus demonstrating reasonable measures have been taken to be compliant with GDPR Article 17 "Right to Erasure":

**Table 2.** Summary of PII breach before implementation and how these have been eliminated.

| Recommendations to be Implemented | Before Using Recommendations | After Using Recommendations |
| --- | --- | --- |
| 1. Allocate a dedicated management laptop configured with trusted hosts, file containing encrypted username and passwords, to connect securely with all devices. | 1. Do not have secure remote connectivity set up to access all devices on each component within the hybrid cloud; therefore, all PII cannot be identified or accessed on every device. In this example scenario, the assigned user only has access to the production database. | 1. The dedicated management laptop has a file containing all the relevant credentials encrypted. These can only be decrypted by the super user (who encrypted them) and only on this device (where they were encrypted) enabling secure remote connectivity to every device on each component within the hybrid cloud, enabling all PII to be identified and located. |

**Table 2.** *Cont.*

| Recommendations to be Implemented | Before Using Recommendations | After Using Recommendations |
|---|---|---|
| 2. Ensure PII is encrypted both at rest and in transit. | 2. An SQL query would be run to identify and report the location of PII in the production database, without using encryption.<br>3. The output from the query would possibly be stored in a document/file that was not encrypted. | 2. PowerShell can encrypt and decrypt so files containing PII will be encrypted in transit as well as at rest.<br>3. Credentials are stored encrypted and only the super user can decrypt these when required.<br>4. Files can be encrypted using an Advanced Encryption Standard (AES) algorithm, so the data retrieved using the script may be encrypted. The use of specified trusted hosts adds to securing access. |
| 3. Ensure event logs and audit trails are in place, to demonstrate reasonable measures were taken to be Right to Erasure ("Right to be Forgotten"). | 4. Do not have audit trails in place that can be used to demonstrate reasonable measures were taken to identify, locate, report and delete PII. This example highlights there is no guarantee which user will be assigned these tasks thus making auditing and event logging harder to trace. | 5. Only one super user will carry out this task; therefore, audit trails and event logs can be used to check the activity of this user and demonstrate reasonable measures were taken to identify, locate, report and delete PII. |
| 4. Create a super user account with administrative authority enabling full access to all devices (physical and virtual) within the hybrid environment. | 5. Do not have authority to access all devices and the devices they have access to, they do not have authority to all PII so unable to identify, locate and report PII. | 6. The super user has the appropriate authority to access all PII on every device so all PII formats can be identified and located within the hybrid cloud environment. |
| 5. Super user must have access to passwords, cryptographic keys etc., names and IP addresses of all devices within the hybrid cloud. | 6. User will not have access to some of the passwords or cryptographic keys etc., and therefore cannot access all PII. | 7. The super user will have access to all passwords, cryptographic keys, names and IP addresses of all devices within the hybrid cloud enabling administrative access to all devices. |
| 6. Identify all the PII that is currently retained. | 7. Retains expired PII.<br>8. Retains more PII than what was/is required for the purpose, thinking they might use it in the future.<br>9. Do not know their data landscape, nor what constitutes PII and as a result upon receipt of a valid request from a data subject to erase their PII, think deleting PII from the production database will suffice. In this example, the assigned user only checks the production database.<br>10. PII stored in other formats were not investigated. | 8. Expired PII is no longer retained as PowerShell can identify and retrieve the location of all PII within the hybrid environment and so can be deleted.<br>9. Whilst identifying all the PII that is currently retained, extra PII may be identified and dealt with, but data minimisation would be carried out under GDPR Article 5 "Principles relating to processing of personal data".<br>10. The data landscape is clear so now all areas containing all formats of PII can be identified and located within the hybrid cloud environment. |

**Table 2.** *Cont.*

| Recommendations to be Implemented | Before Using Recommendations | After Using Recommendations |
|---|---|---|
| 7. Create scripts to identify and locate PII for a data subject and where PII is due to expire, incorporating encryption. | 11. An SQL query would have been run only over the Production database to locate PII, so other databases and data formats throughout the hybrid cloud environment would have been overlooked. <br> 12. Expired PII may be deactivated in some way, but unlikely to have been identified with a view to erasure. | 11. The super user has administrative authority to all PII on every device within the hybrid environment and because of this, the process cannot be fully automated. A manual log on is required to every device and each script run separately. All PII can now be identified for a data subject and PII where the contract has expired. |
| 8. Use an appropriate tool (If using PowerShell, as in this instance, ensure it has been configured on each device). <br> 9. Make sure the Firewall has been configured with the relevant Inbound and Outbound rules for specific ports and IP addresses and Access Control Lists (ACLs) are up and running. | 13. No automation tool or script to identify and report the location of PII for a data subject or PII that has expired. <br><br> 14. No access control lists or firewall rules configured to enable a user or device to access PII on every device. | 12. With all devices configured, PowerShell can be run to locate and retrieve the location of PII for both a data subject and expired PII. <br><br> 13. Access control lists and firewall rules are configured to only allow the dedicated management machine and super user access to PII on every device. |
| 10. Securely delete relevant PII records (remember to empty recycle bins, clear history, remove from backups and archives). | 15. The backup process of nightly, weekly and monthly would erase the PII; however, there could be occasions where the monthly runs late, so PII would not be erased within the time limit of 30 day, which would be a PII breach. <br> 16. Backups and archives may be stored off-site and on tapes <br> 17. PII would only have been deleted from the production database. | 14. The backup process of nightly, weekly and monthly would erase the PII; therefore, procedures are put in place to ensure that the monthly is run on 28th day or nearest weekend to 28th day of the month. <br> 15. Backups and archives may be stored off-site and on tapes. These must all be encrypted and stored in a secure environment and where feasible erased. <br> 16. All PII would be securely deleted from every device within the hybrid cloud. |
| 11. Document steps carried out. | 18. No processes or procedures in place to document. | 17. Steps undertaken would be documented so they can be used to demonstrate that reasonable measures were taken to be compliant with GDPR Article 17, "Right to Erasure". |

## 6. Evaluation

Table 3 shows the structured and semi-structured data was found but some of the unstructured was not. Furthermore, the data that was found was only picked up if the spelling was exact. Regarding the encrypted database, the encrypted password is required to gain access. PowerShell does have special coding for .zip which was not included in the test scripts ran, so .zip is not a true reflection. PII was not picked up from the recycle bin or Snapshots, so further investigation and analysis needs to be conducted on this.

Table 4 shows that all structured, semi-structured and most of the unstructured data was retrieved but .xlsx documents were omitted.

**Table 3.** Summary of tests for retrieval of PII for a data subject.

| PII for Data Subject | | | |
|---|---|---|---|
| **Data Subject** | **LAN** | **PRIVATE** | **PUBLIC** |
| Structured | Yes | Yes | Yes |
| Semi-Structured | Yes | Yes | Yes |
| Unstructured | Yes (.txt) No (.docx, .pdf, .xlsx) | Yes (.txt) No (.docx, .pdf, .xlsx) | Yes (.txt) No (.docx, .pdf, .xlsx) |
| Encrypted | Yes (access database) No (.zip) | n/a | n/a |

**Table 4.** Summary of tests for the retrieval of PII that has expired.

| PII That Has Expired | | | |
|---|---|---|---|
| **Expiration Date** | **LAN** | **PRIVATE** | **PUBLIC** |
| Structured | Yes | Yes | Yes |
| Semi-Structured | Yes | Yes | Yes |
| Unstructured | Yes (.docx, .pdf, .txt) No (.xlsx) | Yes (.docx, .pdf, .txt) No (.xlsx) | Yes (.docx, .pdf, .txt) No (.xlsx) |
| Encrypted | Yes (access database) No (.zip) | n/a | n/a |

Figures 16 and 17 show the file size per year for Server1 and Server2, highlighting the amount of possible obsolete data retained by the organisation.

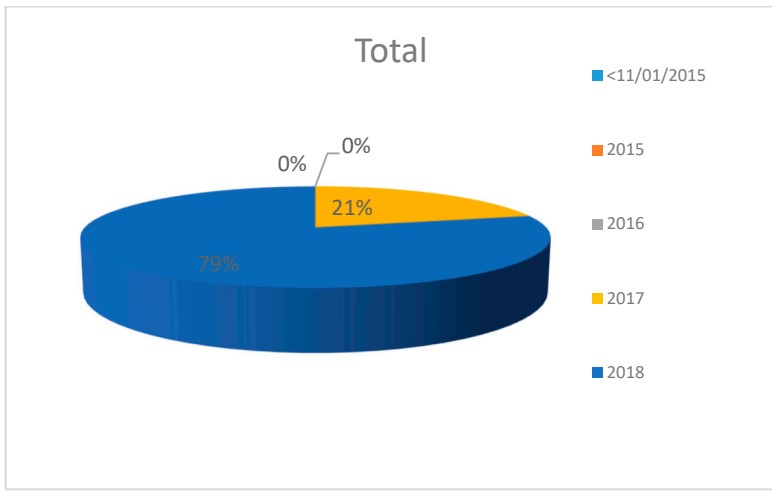

**Figure 16.** Summary of file size by year for Server1.

Figures 18 and 19 show selected document types for Server1 and Server2, highlighting the amount of structured, semi-structured and unstructured data retained and highlights documents that are zipped. To access .zip special coding is required. This is the same for .pdf. So, having an overview of your data landscape can provide some insight to the type of data you need to deal with.

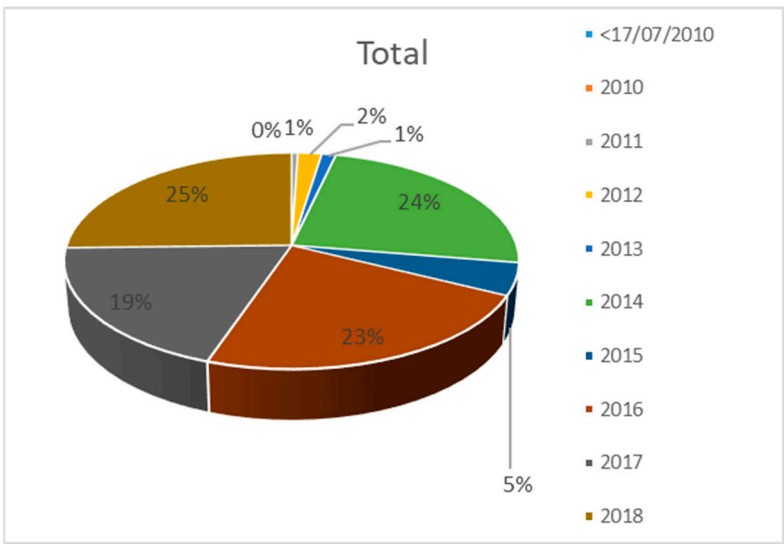

**Figure 17.** Summary of file size by year for Server2.

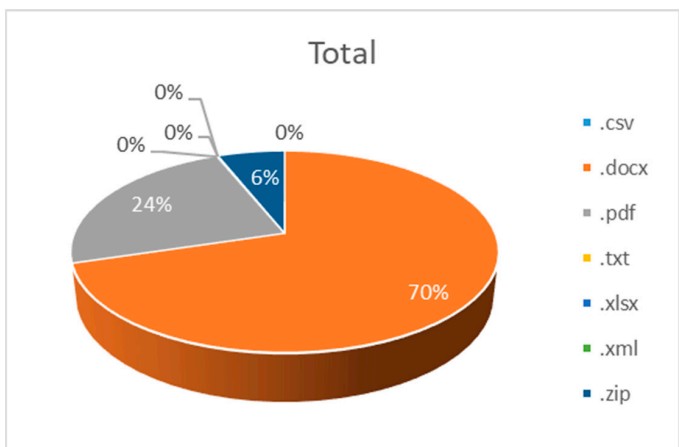

**Figure 18.** Document types held within Server 1.

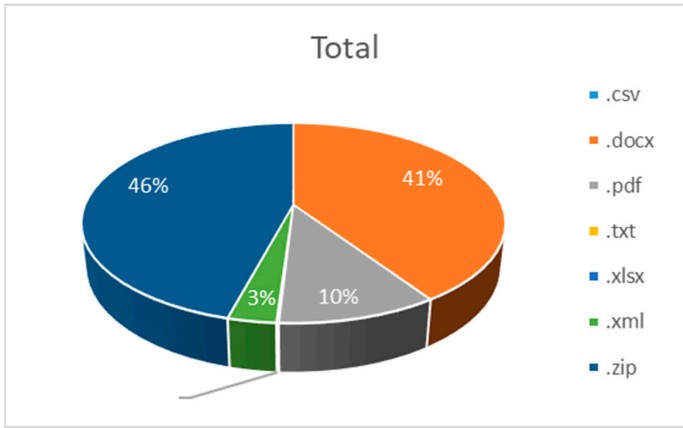

**Figure 19.** Document types held within Server 2.

Visual representation of all files of a data subject named PhilomenaAnnKelly and PHILOMENAANNKELLY1 on Server1 can be seen in Figure 20, and those of the administrator on Server2 can be seen in Figure 21.

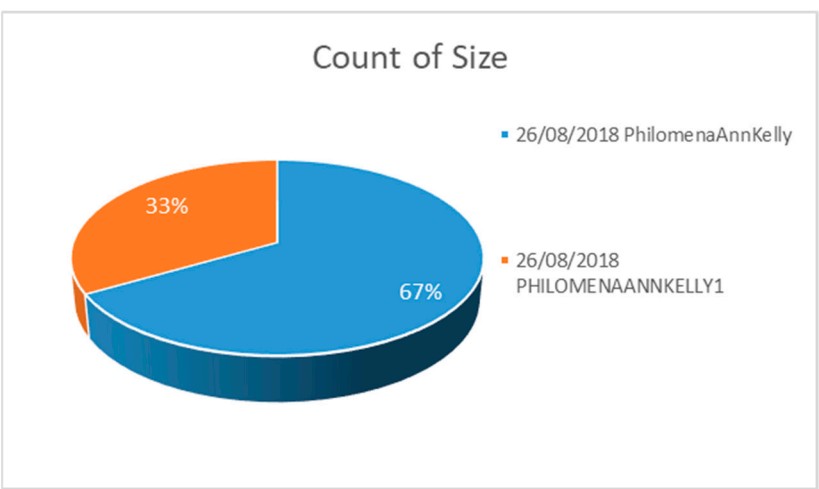

**Figure 20.** Amount of PII held for a specific data subject.

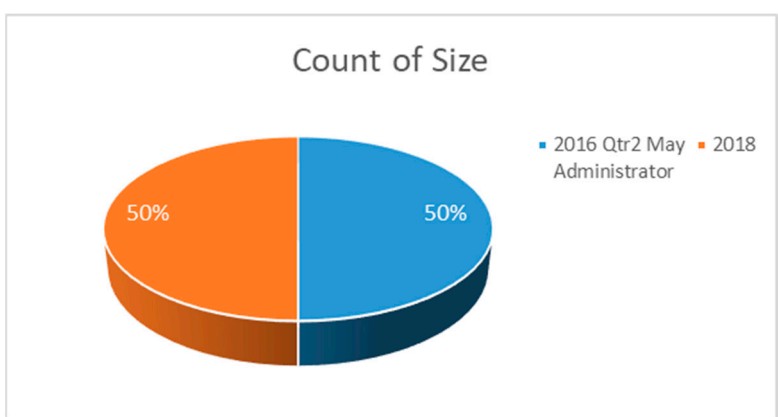

**Figure 21.** Data held on Server 2 for administrator.

GDPR focuses mainly on data privacy, rather than data security, with only 8 out of its 99 Articles dealing explicitly with technology and tools. Article 17, Right to Erasure ("Right to be Forgotten") covers both, as you need to pass through security to access the PII, yet whilst doing this, data privacy must be kept intact both at rest and in transit. Data privacy and data security tend to overlap and become confusing. The industry standards ISO/IEC 27001:2013 and ISO/IEC 27002:2013 offer an Information Security Management System (ISMS), but now are developing the ISO/IEC 27552 to offer a Privacy Information Management System (PIMS). This new standard aims to extend ISO/IEC 27001:2013 and ISO/IEC 27002:2013 to cater for data privacy as well as data security. ISO/IEC 27555 provides a framework for the deletion of PII.

We show that compliance with GDPR's Article 17 Right to Erasure ("Right to be Forgotten") is achievable given a set of recommendations. Listed in Table 5 are the top practical guidelines resulting from the work carried out aimed specifically for existing PII throughout an ad-hoc hybrid cloud environment of a small business to demonstrate reasonable measures for the Article 17 of GDPR, Right to Erasure ("Right to be Forgotten") compliance. This work confirms that compliance with GDPR's Article 17, Right to Erasure ("Right to be Forgotten") is achievable in a hybrid cloud storage environment, if the recommendations are considered.

**Table 5.** Top recommendations.

| | **Top 14 Recommendations to Become Compliant with GDPR Article 17 Right to Erasure ("Right to be Forgotten")** |
|---|---|
| | Dedicated Management Laptop: |
| 1 | Allocate a dedicated management machine configured with trusted hosts, file containing encrypted username and passwords, to connect securely with all devices. |
| | Encryption |
| 2 | Ensure PII is encrypted both at rest and in transit. |
| 3 | Ensure event logs and audit trails are in place, to demonstrate reasonable measures were taken to ensure the Right to Erasure ("Right to be Forgotten"). |
| | Super User: Access and Authority |
| 4 | Create a super user account with administrative authority enabling full access to all PII on every device (physical and virtual) within the hybrid environment. |
| 5 | Have access to passwords, cryptographic keys etc. |
| 6 | Have access to the names and IP addresses of all devices in the hybrid cloud. |
| | Personally Identifiable Information (PII): |
| 7 | Identify all the PII that is currently retained. |
| 8 | Create scripts to identify and locate PII for a data subject and where PII is due to expire, incorporating encryption. |
| | Task Automation and Configuration Management Tool |
| 9 | Use an appropriate tool. (If using PowerShell, as in this instance, ensure it has been configured on each device). |
| | Security: Firewalls |
| 10 | Make sure the Firewall has been configured with the relevant Inbound and Outbound rules for specific ports and IP addresses and Access Control Lists (ACLs) are up and running. |
| | Erasure: |
| 11 | Securely delete relevant PII records (remember to empty recycle bins, clear history and remove from backups and archives, which may be stored off-site and on tape, if feasible). |
| 14 | Document steps carried out. |

## 7. Conclusions

This research set out to examine the GDPR Article 17 "Right to Erasure" and the challenges a small organisation may face whilst implementing this right within a hybrid cloud storage environment. It was found that compliance with GDPR's Article 17 "Right to Erasure" is achievable in a hybrid cloud environment by following a set of recommendations. Our tests, which were carried out on a selection of structured, semi-structured and unstructured data formats stored in a variety of locations within the hybrid cloud environment, illustrate that it is possible to securely connect to each component within a hybrid cloud environment using a non-intrusive free tool like PowerShell and execute scripts to identify, locate and report PII for a data subject and PII that has expired. The structured data format was quite straightforward when identifying and locating a data subject; however, identifying and locating PII that had expired was initially more challenging. This was attributed to the lack of auditing configured on the database; however, once this was implemented it was easy to identify the date using the last date the data was accessed. Implementing a Super User account allowed for event logs and audit trails (if not already in place) to be put in place, both to monitor this privileged account and to be used to demonstrate compliance. Since the Super User has such high privilege within the hybrid cloud, it is considered best practice that this process is not fully automated. Instead, it is mandatory that manual logins/authentication are carried out. Although laborious, this is eased by encrypting the usernames and passwords, storing them within an encrypted file that only can be accessed by the user that created it and from the machine in which it was created. Full administrative access is required to enable full access to all areas within the hybrid as this is required to avoid access being denied. Some devices needed a device

name whilst others required an IP address, but either way they must be added to the trusted host on the management laptop, and the management laptop must be added as a trusted host on each device. Passwords and cryptographic keys are required for scripts to access encrypted folders, files and databases. All connections to the relevant devices must be closed as soon as the task is complete and execution policy set back to restricted. It is crucial for organisations to identify all the PII currently retained, and it is best to start with the high-risk areas of the business such as processes dealing with payments, or processes involving third parties. All scripts used must incorporate encryption and PowerShell enables this. PowerShell must be configured on each device to enable remoting, but this must be set back to restricted once task is complete. This is another security feature of PowerShell. Firewalls must be configured to secure specific ports and only allow the management IP address. ACL must be set to allow Super User through.

Whilst is it not possible to guarantee complete assurance that all PII has been identified and removed, these recommendations can be referred to by a small organisation running a hybrid cloud to demonstrate that they have taken reasonable measures to be compliant with the "Right to Erasure". Therefore, this research has identified that compliance with GDPR's Article 17 "Right to Erasure" is achievable in a hybrid cloud storage environment.

**Author Contributions:** M.K. conducted the primary research. E.F. designed the framework and supervised the research. K.C. provided consultancy and helped shaped the final paper. All authors have read and agreed to the published version of the manuscript.

**Funding:** This research received no external funding.

**Institutional Review Board Statement:** Not applicable.

**Informed Consent Statement:** Not applicable.

**Data Availability Statement:** Data presented in this study relating to GDPR are available in summarized form on request from the corresponding author. Other data presented in this study are available on request from the corresponding author.

**Conflicts of Interest:** The authors declare no conflict of interest.

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
