# Peer review of "How to Achieve Compliance with GDPR Article 17 in a Hybrid Cloud Environment"

_sci, doi:10.3390/sci3010003_

Round 1

Reviewer 1 Report

The paper investigates the fulfillment possibilities of the Right to be Forgotten principle of the EU GDPR in hybrid cloud environments. It also provides guidelines with a list of recommendations to be followed for compliance.

The paper is basically well written and readable. The abstract and introduction well positions the performed research, though the exact contributions may be further highlighted. What does "100% of time" mean in the abstract?

Section 2 provides a good overview of the main contents of GDPR (though it has been summarized by many works before). Figure 1 has a typo: panalties -> penalties.

Section 3 introduces the cloud computing technology, and characterizes hybrid clouds.

Section 4 introduces a sample test hybrid cloud environment to be used for exemplifying data retrieval and erasure cases. Though the approach manages to highlight certain cases, it is not broad and detailed enough to cover most (even not all) cases. An earlier work on data protection compliance in multi-clouds related to this paper is recommended for broadening this proposal: R. Garg, Sz. Varadi, A. Kertesz, Legal Considerations of IoT Applications in Fog and Cloud Environments. PDP'19, pp. 193-198, 2019. DOI: 10.1109/EMPDP.2019.8671620.

At some parts the test-bed environment has some unnecessary restrictions (e.g. the use of an SQL database, or a certain virtualization technique such as VMWare). Nevertheless, should a distributed file storage system be used in the system, data retrieval and erasure might be more complicated (specially in cases VMs were involved). It is also unclear from the description, if the test-bed was installed and set up in real world, or not.

Section 5 evaluates certain test cases on the test-bed to demonstrate the inner workings, or steps to be taken for performing personal data erasure. Concerning Figure 5, I cannot see why a user (e.g. 1) would not be able to access servers at different locations at the same time. What does "expired PII" mean?

Concerning the test case in general, I cannot see why an SME would let PII of users spread over servers unguided, so special scripts would be needed to mine all data out. It would be logical to group data of a user to a folder, or to certain records with a corresponding ID/key, instead - which would make data retrieval trivial. Concerning public cloud usage, it would also be rational to use temporal VMs there, and after a certain task (e.g. data processing) is over, the whole VM is decommissioned, so no data remain in a public cloud.

Section 5.3 discusses the proposed recommendations to fulfill GDPR requirements in the given scenario, and compares the original and revised management processes. Section 6 further analyses the test scenario, draws conclusion and summarizes generalized recommendations (to Table 5).

Section 7 is too long, and restates some earlier text. It should be more condensed.

Overall, I found this work interesting and beneficial, therefore it is worth to be published. Nevertheless, the considered test-bed and scenario is very restrictive, and I do not think many SMEs have such a setup, and use similar cloud setups. The case itself is also quite static: real cloud properties are not addressed and exploited - almost the entire test-bed environment could be set up without virtualization, using remote physical servers.

Reviewer 2 Report

The study aims to demonstrate compliance with GDPR’s Article 17 Right to Erasure (‘Right to be Forgotten’) is achievable in a Hybrid cloud environment by following a list of recommendations. The study is interesting and meaningful. However, I have several questions:

  1. The full name of ‘General Data Protection Regulation’ is redundant. The full name of ‘OWASP’ abbreviation is missing. Please check other abbreviations.
  2. The study method is unclear. Please strengthen this part to make the study have more theoretical depth. Section 4.1, 4.2, and 4.3 are quite technical instead of methodological manner.